# Development and Optimization of Djulis Sourdough Bread Fermented by Lactic Acid Bacteria for Antioxidant Capacity

**DOI:** 10.3390/molecules26185658

**Published:** 2021-09-17

**Authors:** Hung-Yueh Chen, Chang-Wei Hsieh, Pin-Cheng Chen, Shin-Pin Lin, Ya-Fen Lin, Kuan-Chen Cheng

**Affiliations:** 1Institute of Food Science and Technology, College of Bioresources and Agriculture, National Taiwan University, Taipei 10617, Taiwan; yuwhyueh@gmail.com (H.-Y.C.); sdf1230648@yahoo.com.tw (P.-C.C.); 2Department of Food Science and Biotechnology, National Chung Hsing University, Taichung 402, Taiwan; welson@nchu.edu.tw; 3Department of Medical Research, China Medical University Hospital, China Medical University, 91, Hsueh-Shih Road, Taichung 404, Taiwan; 4School of Food Safety, Taipei Medical University, Taipei 11031, Taiwan; splin0330@tmu.edu.tw; 5Department of Agronomy, National Taiwan University, Taipei 10617, Taiwan; 6Institute of Biotechnology, College of Bioresources and Agriculture, National Taiwan University, Taipei 10617, Taiwan; 7Department of Optometry, Asia University, 500, Lioufeng Rd., Wufeng, Taichung 41354, Taiwan

**Keywords:** *Chenopodium formosanum*, sourdough, lactic acid bacteria, antioxidant

## Abstract

This study developed a nutritionally valuable product with bioactive activity that improves the quality of bread. Djulis (*Chenopodium formosanum*), a native plant of Taiwan, was fermented using 23 different lactic acid bacteria strains. *Lactobacillus casei* BCRC10697 was identified as the ideal strain for fermentation, as it lowered the pH value of samples to 4.6 and demonstrated proteolysis ability 1.88 times higher than controls after 24 h of fermentation. Response surface methodology was adopted to optimize the djulis fermentation conditions for trolox equivalent antioxidant capacity (TEAC). The optimal conditions were a temperature of 33.5 °C, fructose content of 7.7%, and dough yield of 332.8, which yielded a TEAC at 6.82 mmol/kg. A 63% increase in TEAC and 20% increase in DPPH were observed when compared with unfermented djulis. Subsequently, the fermented djulis was used in different proportions as a substitute for wheat flour to make bread. The total phenolic and flavonoid compounds were 4.23 mg GAE/g and 3.46 mg QE/g, marking respective increases of 18% and 40% when the djulis was added. Texture analysis revealed that adding djulis increased the hardness and chewiness of sourdough breads. It also extended their shelf life by approximately 2 days. Thus, adding djulis to sourdough can enhance the functionality of breads and may provide a potential basis for developing djulis-based functional food.

## 1. Introduction

Bread is widely considered a staple food in the human diet [1]. As a result of greater health awareness, the quality of functional and nutritional breads is often improved via additional food processing or the adjunction of other materials. For instance, adding cowpea or quinoa enhances the protein and dietary fiber content of bread, provides essential amino acids for the body, and promotes gastrointestinal motility [2,3]. Adding quinoa in particular also lowers blood lipids and increases antioxidant capability [4]. Moreover, alternative sources of cereals, especially pseudo-grains, improve the nutritional and functional components of traditional wheat bread [5], including increasing the content of digestible protein [6], reducing allergies by making gluten-free bread [7], and enhancing antioxidant capacity [8].

Djulis (*Chenopodium formosanum* Koidz) is a pseudocereal plant and native species of Taiwan [9]. In addition to being high in protein (approximately 14.4%), dietary fiber, and minerals, it is also abundant in essential amino acids, potassium, calcium, magnesium zinc, and trace elements such as selenium and germanium. The bioactive components of djulis exhibit antioxidant, antidiabetic, anti-inflammation, and immune regulation effects [10]. Its functional components mainly include phenolic acids, flavonoids, triterpenes, sterols, and nitrogen-containing compounds. Djulis contains two phenolic acids. The first is a group of benzoic acid analogs, which are abundant in the leaves and seeds. The bioactivities of these analogs include antibacterial, antioxidant, and so on. The second is a group of cinnamic acid analogs, which are primarily stored in djulis as conjugated phenols. Their bioactivities include antioxidant, antibacterial, and antiapoptotic activity and the amelioration of diabetes symptoms [11]. Betalain present in djulis also boosts antioxidant capability, and this was found to be more powerful than α-tocopherol in inhibiting lipid oxidation and the proliferation of melanoma [12].

With developments in microbial technology, increasingly more studies have confirmed that baking sourdough fermented by lactic acid bacteria (LAB) improves the nutritional content and textural quality of the bread [13]. Traditional sourdough fermentation involves a mixture of flour and water being fermented by ingenuous LAB and yeast. In the ecosystem of sourdough, LAB is the dominant microorganism and has a considerable influence over the characteristics of the sourdough. LAB generally refers to Gram-positive bacteria grown in a facultative anaerobic or absolutely anaerobic environment without spore production. The metabolites of the carbohydrates are mainly lactic acid [14]. LAB are used in popular probiotic products containing live bacteria that provide health benefits to the host [15]. The multiple bioactivities and functions of LAB are beneficial to humans, such as in regulating immune function, reducing lactose intolerance symptoms, and decreasing blood pressure and cholesterol levels [16,17] and by providing antioxidation [18] and anticolorectal cancer effects [19]. LAB also exhibit several activities that benefit food fermentation, such as reducing pH levels to extend the shelf life of food and producing extracellular polysaccharides to improve product texture. By producing oligosaccharides and vitamin E, releasing active peptides, and degrading phytic acid and stachyose, LAB fermentation also enhances nutritional value [20].

Response surface methodology (RSM), which differs from the one-factor-at-a-time approach [21], is a method for predicting optimal production and for combining experimental design and statistical analysis [22,23]. RSM is employed in applications such as enzyme immobilization [24], bioethanol production [25], and winemaking [26].

In the present study, djulis was fermented using LAB, and RSM was employed to achieve the optimal antioxidant capacity, total phenols content, and total flavonoid content. The optimally fermented djulis was further processed into bread to improve its nutritional content and texture.

## 2. Results and Discussions

### 2.1. Strain Selection for Djulis Fermentation

Due to increasing health concerns, food processing methods that improve the quality, nutritional content, and bioactive compounds of food are a crucial research topic. Sourdough fermentation in particular has received considerable research interest. Studies have confirmed that sourdough fermentation by LAB enhances bioactive capacity and improves texture. The nutritional content of djulis was rich in protein, lipid, ash, and essential amino acids; djulis was particularly rich in amino acids with sulfur groups and lysine, and it was one of the few vegetable proteins that provided complete essential amino acids [27]. During djulis fermentation, the lactic acid bacteria (LAB) promoted the acidification of dough and the hydrolysis of protein [28], and the decrease in pH value extended the shelf life of bread [29], while the produced protease and reductase weakened the gluten structure and made the bread texture softer [30]. To select a suitable LAB strain for djulis sourdough fermentation in the present study, 23 LAB strains were screened based on their pH level and proteolysis activity after 24 h of fermentation. Most of the LAB strains decreased the pH value to 4.6 (Figure 1A) and boosted the proteolysis activity 1.3 times compared with the controls (Figure 1B). Among the 23 strains, *Lactobacillus casei* BCRC10697 exhibited the highest relative proteolysis activity (1.88 times relative to the controls).

The decision to determine strain selection by pH value and proteolysis activity was based on research by Bustos et al., who showed that chia seeds fermented by LAB exhibited an increase in total phenols, enhanced antioxidant capacity, and improved texture [31]. Because of the higher proteolysis activity of LAB, the better utilization of the fermentation substrate, and the more active compounds (e.g., peptides), LAB fermentation improves antioxidant capacity and protein digestibility. Moreover, the peptides contain fragrance compounds that influence sensory quality [32]. Its ability to lower pH levels, which is influenced by lactic acid, acetic acid, and carbon dioxide produced during fermentation, is another important consideration in sourdough fermentation. Lower pH levels not only provide a unique flavor, but they also effectively increase the shelf life of bread [33].

The growth curve for *L. casei* BCRC10697 shows that the cell count (Figure 1C) and ABTS radical scavenging activity (Figure 1D) peaked after 24 h of fermentation. Therefore, this fermentation time was adopted in subsequent experiments with *L. casei* BCRC10697, as it indicated the greatest fermentation potential for functional sourdough development.

### 2.2. Optimization of Sourdough Fermentation

According to Corsetti et al., factors affecting the overall quality of sourdough can be divided into internal and external factors. Internal factors include the amount of inoculation, temperature, dough yield, oxygen concentration, and fermentation time. External factors include carbon sources, nitrogen sources, minerals, and lipids [34]. The present study considered carbon sources, nitrogen sources, fermentation temperature, initial pH level, mineral concentrations, and dough yield as factors for Box–Behnken design-response surface methodology (BBD-RSM), which was employed to optimize the trolox equivalent antioxidant capacity (TEAC).

The final selection parameters from the BBD-RSM analysis were the fermentation temperature (30 °C–40 °C), dough yield (250–450), and fructose concentration (6–10%). Table 1 shows the experimental design and determined TEAC. The quadratic regression equation (Equation (1)) of the three-dimensional RSM model, which was generated from 20 results and Equation (2), is as follows:*Y = −*0.0193*X*_1_^2^ − 0.0737*X*_2_^2^ − 0.0001*X*_3_^2^ − 0.0056*X*_1_*X*_2_ − 0.0002*X*_2_*X*_3_ + 1.3495*X*_1_ + 1.3977*X*_2_ + 0.0482*X*_3_ − 29.0161(1)
where *Y* is the TEAC (mmol/kg), *X*_1_ is the fermentation temperature (°C), *X*_2_ is the fructose concentration, and *X*_3_ is the dough yield. Relationships between TEAC and the three parameters are expressed as surface and contour plots in Figure 2. As shown in Table 2, the lack of fit was nonsignificant (*p* = 0.425 > 0.05), indicating the RMS accurately predicted TEAC. Moreover, the three factors differed significantly in the first and second terms (*p* < 0.05), and no interaction was observed between factors (*p* > 0.05). The difference between *R* and the adjusted-*R* (R_adj_) of the RSM model was 9.48% < 20%, indicating that the three factors contributed to the optimal equation [35]. The R_adj_ of >0.8 also met conditions applicable to the second-order model [36]. According to Equation (1), the optimal TEAC was 7.12 mmol/kg under the optimal conditions, where the fermentation temperature, fructose concentration, and dough yield were 33.54 °C, 7.7%, and 332.83, respectively. Djulis sourdough was thus prepared under these conditions. The resulting TEAC was 6.82 ± 0.09 mmol trolox/kg, which approximated the theoretical value.

### 2.3. Determination of Antioxidant Capacity

To determine the antioxidant capacity of the sourdough, we compared the unfermented grains, original products (fermentation temperature = 37 °C; dough yield = 300), and optimal products on the basis of antioxidant capacity, TEAC_DPPH_, TEAC_ABTS_, total phenolic compounds, total flavonoid compounds, and total peptides (Table 3). Relative to the unfermented grains, significant increases in TEAC_ABTS_ were observed in the original products (1.5 times) and optimal products (1.66 times). Notably, these results were better than those reported for fermented chia seed sourdough [37] and fermented wheat [38]. For TEAC_DPPH_, the optimal products were significantly better than the unfermented grains (1.2 times) and original products (1.1 times). The optimal products also exhibited higher antioxidant capacity compared with cowpea bean dough fermented by *Lactobacillus plantarum* [39].

The optimal products enhanced the total phenolic compounds and total flavonoid compounds by 49% and 17%, respectively. This is similar to results reported by previous research [40]. The acid produced by LAB fermentation accelerates the extraction of phenolic and flavonoid compounds, and the lipolytic enzyme secreted by LAB could assist with the hydrolysis of combined phenolic and flavonoid compounds [41].

After fermentation, the total peptide content of the optimal products was improved approximately 2.14 times compared with the unfermented grains, which was due to the proteolysis enzymes secreted from the LAB degrading proteins into short-chain peptides. In the initial state of fermentation, the correlation between the peptide content and antioxidant capacity is highly positive; by contrast, that in the post state of fermentation is based on the phenolic compound content [42].

To determine the antioxidant capacity in bread, samples of wheat bread (WB), djulis bread (DB), 20% djulis bread (DSB), and 25% DSB were prepared and analyzed. The results are shown in Table 4. The 20% DSB and 25% DSB did not differ significantly in TEAC_ABTS_, but they improved 2 times relative to WB and 15% relative to DB. Moreover, the TEAC_DPPH_ of 20% DSB and 25% DSB improved considerably from WB and increased 22% relative to DB. Although the antioxidant capacity was reduced after baking, an improvement was observed after djulis was added to the sourdough. The results are similar to those reported by Rizzello et al., who reported that djulis increased the antioxidant capacity by 9% [40].

The total phenolic compound content and total flavonoid compound content of the DSB were greatly improved relative to the WB and DB, due the increase in antioxidant capacity [43]. The total peptide content was enhanced after the addition of djulis and sourdough fermentation. Previous research has reported an association between the enhancement of total peptide content with antioxidant capacity [44] and antibacterial ability in extending product shelf life [45].

### 2.4. Analysis of Bread Properties

The specific volume and structure of bread is an important factor when evaluating its exterior quality. The rich fiber and starch in djulis compete with gluten for water and turns gluten into glutenin by destroying the β-turn to form a β-sheet structure. In addition, the increase in polyphenols also destroys the β-turn of the gluten and forms a β-sheet structure. The continuous β-turn structure in the gluten and adjacent short chain α-helix form a β-spiral structure, making the gluten viscoelastic. Thus, the loss of the elastic β-spiral structure causes gluten to accumulate, which decreases the specific volume of the bread [46].

In the present research, a significant decrease was observed in the specific volume of the bread produced with djulis added (Table 5). Moreover, the decrease in the DSB was significantly more compared with that in the DB, and this was attributed to the increase in phenolic compounds after fermentation. The bread structure was evaluated according to the gas area in a cross-section of the bread (area%) and the average bubble size. Generally, a larger area indicates that the texture of the bread is more elastic and softer [47]. Our results show that this area became smaller after djulis was added; it was larger in the DSB than in the DB, although the specific volume in the DSB was lower. Although the DSB produced more pores, the gas retention capacity during the baking process was not as favorable as that in the unfermented djulis group; this was due to damage to the gluten structure. Hardness and springiness increased with the addition ratio, due to the proline residue in the gluten being hydrolyzed by the protease secreted from the LAB. Additionally, the gluten was weakened by the lactic acid and acetic acid; this resulted in the formation of shorter and harder gluten, making the texture of bread more stable and elongated but with less elasticity. In conclusion, the specific volume of bread was decreased, and this increased the hardness and springiness [48].

Table 5 shows the differences in the general nutritional components of the different bread groups. The moisture of the DSB was lower than that of the WB and DB because the djulis addition that replaced the gluten content of the dough reduced the water retention of the bread [49]. According to Mantzourani et al., the recommended moisture for bread is 31–35%. Therefore, adding too much djulis had a negative impact on the bread quality. In addition, the protein and ash content increased after djulis was added, mainly because djulis has richer protein content and ash content compared with wheat. Moreover, the increase in protein and lipid content was probably related to the fatty acids, proteases, and amino acids produced by the LAB during fermentation [50].

### 2.5. Sensory Evaluation of Bread

The appearance, flavor, taste, texture, and overall acceptability of the bread were given scores ranging from 1 to 9 points, where 1 = extremely dislike and 9 = extremely like (Table 6). The results corroborated our observation that the unfermented djulis of the DB groups would adversely affect the acceptability. The 20% DSB groups were more acceptable than the DB, but they were slightly less acceptable than the WB groups. To make the DSB more acceptable to consumers, DSB was also prepared with cocoa powder and dried cranberry (DSBC). The DSBC groups outperformed the other groups in the appearance, flavor, taste, texture, and overall acceptability, and this was attributed to the cocoa and cranberry covering the astringency of the djulis.

Sensory evaluation was performed through a principal component analysis. As shown in Figure 3, the sum of PC1 and PC2 was greater than 80%, indicating that these variables were sufficiently representative [51]. The four sample groups fell into four different quadrants, with obviously different characteristics among the four groups. The DB groups on the right side of the figure were associated with being more bitter, astringent, and musty and with having undesirable aftertastes. After fermentation, the 20% DSB groups reduced the bad flavor, and the 20% DSBC groups also reduced the bad flavor while improving the acceptability; these effects were due to a reduction in saponin content and bitterness [52]. The 20% DSB and 20% DSBC groups were in the third and fourth quadrants, with the characteristics of the sour and acetic flavor of the sourdough, while the WB and DB groups had a more obvious yeasty flavor.

## 3. Materials and Methods

### 3.1. Materials

Djulis (*Chenopodium formosanum*) was purchased from Taitung country farmers’ association (Taitung, Taiwan). Wheat flour was purchased from Yifeng food company (New Taipei City, Taiwan). MRS broth was purchased from Hardy Diagnostics (Santa Maria, CA, USA). Peptone and Bacto agar were purchased from Bioshop (Burlington, ON, Canada). Glucose, sucrose, lactose, magnesium sulfate, sodium phosphate dibasic, potassium phosphate dibasic 2-mercaptoethanol, sodium dodecyl sulfate, sodium tetraborate, monosodium dihydrogen orthophosphate, sodium phosphate dibasic, 2,2-diphenyl-1-picrylhydrazyl, sodium chloride, sodium carbonate, potassium persulfate, hydrochloric acid, 2,2′-azino-bis (3-ethylbenzothiazoline-6-sulphonic acid), trolox (6-hydroxy-2,5,7,8-tetramethylchromane-2-carboxylic acid) were purchased from Sigma-Aldrich (St. Louis, MO, USA). *o*-Phthaldialdehyde was purchased from Merck (Darmstadt, Germany).

### 3.2. Microorganisms and Medium

Lactobacillus paracasei BCRC14023, Lactococcus lactis BCRC12315, Lactobacillus helveticus BCRC14092, Lactobacillus rhamnosus BCRC10940, Lactobacillus johnsonii BCRC17474, Lactobacillus brevis BCRC12247, Lactobacillus delbrueckii BCRC12195, Lactobacillus gasseri BCRC14619, Lactobacillus reuteri BCRC14625, Lactobacillus helveticus BCRC12936, Lactobacillus delbrueckii BCRC14009, Bifidobacterium infantisI BCRC14633, Bifidobacterium longum BCRC14602, Bifidobacterium adolescentis BCRC14606, Bifidobacterium bifidum BCRC14615, Bifidobacterium longum BCRC14634, Bifidobacterium breve BCRC11846, Lactobacillus rhamnosus BCRC16000, Lactobacillus delbrueckii BCRC10696, Lactobacillus plantarum BCRC11697, Lactobacillus acidophilus BCRC14079, Streptococcus thermophilus BCRC14085, Lactobacillus casei BCRC10697 were purchased from Bioresource Collection and Research Center (BCRC, Hsinchu city, Taiwan) and cultured in MRS medium for routine use (Sigma-Aldrich, St. Louis, MO, USA). Bacteria were stored in MRS broth with 20% glycerol at −80 °C for long-term storage. For the activation of bacteria, all strains were thawed at room temperature and then inoculated 1% of bacteria (*v/v*) into the MRS broth and cultured at a constant temperature at 37 °C for 48 h and sub-cultured twice a week.

### 3.3. Sourdough Bread Production

Djulis was grounded into powder and filtered through a 40 mesh screen and stored at 4 °C until use. Djulis powder was sterilized in a vertical autoclave at 121 °C for 30 min, then combined with 0.5 times the amount of ddH_2_O (*w/v*) for making the sourdough, of which the dough yield was 300. Samples were inoculated with 2% LAB and fermented at 37 °C for 48 h. To determine the best condition for LAB fermentation, every LAB was cultured for 24 h, and the biomass, the pH value, and the protease activity were monitored. The selection of LAB was based on the protease activity. The sourdough bread production was modified according to the method reported by Bartkiene et al. [53]. The material ratio was calculated by baking percentage (total weight of flour and djulis as 100%). According to Table 7, the ingredients were combined in a bread machine for first stage fermentation at 30 °C for 50 min, followed by the shaping, cutting, and removing the gas of the dough. At the second stage fermentation, samples were cultured at 30 °C, 80–85% relative humidity for 30 min, and baked at 200 °C for 20 min.

### 3.4. Optimization for Djulis Fermentation by the Response Surface Methodology (RSM)

The optimal conditions for LAB fermentation had determined by using Box–Behnken design-response surface methodology (BBD-RSM). Effects of the cultivation temperature (*X*_1_) at 30 °C–40 °C, the fructose concentration (*X*_2_) in 6–10%, and the dough yield (*X*_3_) in 250–350 on the trolox equivalent antioxidant capacity (TEAC) were investigated. A three-dimensional RSM model was designed after 15 experiments and predicted the optimal condition (Table 1) using Minitab software (version 16, Minitab Inc., State College, PA, USA) to perform regression analysis and surface plotting, which was conducted to predict the optimal value of the three variables. The design of the three-dimensional RSM model was based on the following quadratic equation:(2)$$Y=\beta_{0}+\overset{3}{\underset{i=1}{\sum }}\beta_{i}X_{i}+\overset{3}{\underset{i=1}{\sum }}\beta_{ii}X_{i}^{2}+\overset{3}{\underset{i=1}{\sum }}\overset{3}{\underset{j=1}{\sum }}\beta_{ij}X_{i}X_{j}$$
where *Y*, *β*_0_, *β_i_*, *β_ii_*, *β_ij_*, *X_i_*, and *X_j_* were the response variable, model constant, linear coefficient, quadratic coefficient, interaction coefficient, and independent variables, respectively.

### 3.5. Determination of Protease Activity and Total Peptide Content

The preparation of *o*-phthaldialdehyde (OPA) solution was modified according to the method reported by Church et al. [54]. An amount of 40 mg of OPA was dissolved in 1 mL methanol and well mixed with 25 mL of 100 mM sodium tetraborate, 2.5 mL of 20% SDS, and 100 μL 2-mercaptoethanol, with the addition of ddH_2_O to result in a final volume of 50 mL. The fermented djulis samples were extracted using 80 mM phosphate-buffered solution at pH 8.5 at room temperature for 1 h, and centrifuged at 7000× *g* for 30 min. The supernatant (50 μL) was taken and reacted with 1 mL OPA solution at room temperature for 2 min, and its OD_340_ was measured. The standard curve of the total peptide content was measured with 0–1 mg/mL leucine, and the equivalent of the total peptide content was expressed by leucine concentration. The protease activity was the relative activity obtained by dividing the peptide content of samples of the control group.

### 3.6. Determination of Antioxidant Activity

Determination of the DPPH and ABTS scavenging activity modified the method reported by Hu et al. [55]. Fermented djulis samples were freeze-dried and re-dissolved in 80% methanol solution.

For the determination of DPPH scavenging activity, 50 μL of sample was added into 150 μL, 100 mM DPPH solution for reaction in the dark for 30 min at room temperature, and its absorbance at OD_517_ was measured using a Multiskan GO microplate spectrophotometer (Thermo Fisher Scientific, Abingdon, UK). The DPPH scavenging activity was calculated by the following quadratic equation (Equation (3)):(3)DPPH scavenging activity (%)=[1−(A1−A2)/A0]×100
where *A*0, *A*1, and *A*2 were the absorbance of DPPH only, samples with DPPH, and the background of samples, respectively.

For the ABTS scavenging activity, two solutions, 7 mM ABTS solution and 2.45 mM potassium persulfate (K_2_S_2_O_8_) were mixed in a ratio of 1:1, and reacted for 12 h until the solution turned to blue-green with to the presence of ABTS^+^. After the absorbance of ABTS solution was adjusted to 0.7 ± 0.02 at OD_734_ by dilution with 0.2 M phosphate buffer solution (PBS, pH 7.4), 20 μL of sample was added to 180 μL ABTS solution and reacted in the dark for 6 min. Sample was measured for its absorbance at OD_734_ using a microplate spectrophotometer. The ABTS scavenging activity was calculated by the following quadratic equation (Equation (4)):
(4)ABTS scavenging activity (%)=(As−Ac) / Ac×100where *Ac* and *As* were the absorbance of ABTS only and samples with ABTS, respectively.

The DPPH and ABTS scavenging activities were represented by trolox equivalent antioxidant capacity (TEAC) (mmol/mL) calculated by the above methods.

### 3.7. Determination of Active Compounds

The method for the total phenolic compounds determination in this study was adopted from Wu et al. [25], with slight modification. Briefly, 100 mg freeze-dried samples were extracted by 1 mL of 50% methanol and shaken at 250 rpm for 1 h, which was then centrifuged at 2100× *g* at 4 °C for 10 min, and the supernatant was taken for filtration through a 0.45 mm filter. After filtration, samples were added with 1 mL acetone/water solution (70:30, *v/v*) for secondary extraction and shaken at 250 rpm for 1 h, which was subsequently centrifuged at 2100 g at 4 °C for 10 min to elicit the supernatant for filtration through a 0.45 mm filter to obtain the free phenolics fraction. The final extraction was reached by adding 1 mL of methanol/H_2_SO_4_ solution (90:10, *v/v*) and was heated in a water bath at 85 °C for 10 h. The samples were centrifuged at 2100× *g* at 4 °C for 10 min to elicit the supernatant for filtration through a 0.45 mm filter to obtain the bound phenolics fraction. Every fraction was mixed and combined with the Folin–Ciocalteu reagent in a 1:1 ratio. After standing for 3 min, 10% aqueous sodium carbonate (Na_2_CO_3_) solution was added to the mixture. After allowing the solution to stand for 1 h, samples were measured at OD_750_. The total phenol compounds content was calculated according to the calibration curve of gallic acid, which was expressed as milligrams of gallic acid per gram of djulis powder (mg GAE/g DP).

Determination of the total flavonoid compounds content was adopted from Wu et al. [25] with slight modification. Briefly, 0.5 mL of methanol extraction sample was mixed with 0.1 mL of 10% aluminum chloride and 0.1 mL of 1 M potassium acetate. After standing for 30 min, samples were measured at OD_415_, and the total flavonoid compounds content was expressed in milligrams of quercetin per gram of djulis powder (mg QE/g dm).

### 3.8. Property Analysis (Texture Analysis, Bread Specific Volume Analysis, Hole Analysis)

The method for texture analysis was modified from Rizzello et al. [40]. Briefly, samples were analyzed using a texture analyzer (TA XT plus) with the P/100 probe (the diameter was 100 mm) (Texture Technologies, Hamilton, MA, USA), and the results were analyzed by the texture profile analysis model. The measured parameters of the early-stage speed, mid-stage speed, and late-stage speed were 1 mm/s, 2 mm/s, and 10 mm/s, respectively. First, samples were pressed down by 10 cm, and subsequently pressed down after 5 s. The hardness, the brittleness, the resilience, and the chewiness were determined.

The bread specific-volume analysis was measured based on the seed substitution method of AACC 10–14.01 [56], in which the rapeseed was full in a fixed container, after which the volume of the rapeseed was measured. After pouring out the rapeseed, the samples and the rapeseed were placed into the container. When the container was full, the remaining rapeseed (W) was weighed and divided by the density of the rapeseed to obtain the volume of samples (Equation (5)).
The volume of samples = W/0.651 (the density of rapeseed).(5)

The hole analysis was determined by ImageJ (version 1.53k, National Institutes of Health, Bethesda, MA, USA), after samples were baked and cooled down at room temperature for 2 h. Samples were cut in half, and the cross-sectional view of samples was taken at a fixed distance for analysis.

### 3.9. Nutritional Analysis (Moisture, Ash Content, Crude Fat, Crude Protein, Carbohydrate)

Determination of the moisture, carbohydrate, crude protein, crude fat, and ash content of fermented djulis samples were adopted according to the methods reported by Wu et al., with slight modification [40]. Briefly, moisture was measured by oven-drying at 105 °C; total carbohydrate content was measured by 100 minus the sum of moisture, crude protein, crude fat, and ash content. Crude protein content was determined using the Kjeldahl method, based on the nitrogen level and multiplied by 5.7. Crude fat content was determined using Soxhlet extraction by petroleum ether. Ash content was analyzed by furnacing at 600 °C overnight.

### 3.10. Sensory Evaluation

The sensory evaluation was carried out based on the nine-point method of consumer preference analysis [57]. Different groups of samples were evaluated by 35 people, and the samples were randomly numbered.

The candidates were wheat bread (WB), djulis bread (DB), djulis sourdough bread (20% DSB), and chocolate djulis sourdough bread (20% DSBC). During the organoleptic experiment, the participants rinsed with mineral water or edible soda biscuits each time when the sample was changed to remove the residual taste in the mouth. The characteristics of samples were evaluated, including assessments of the appearance, the flavor, and the taste; a descriptive analysis; and an overall likeableness score (1–9 points), 1 point was the most disliked and 9 point was extremely liked. The organoleptic evaluation list is shown in Appendix A.

### 3.11. Statistical Analysis

The results of all experiments are expressed as mean ± standard deviation, and experiments were performed at least in triplicate. The data were analyzed using Prism, Minitab, Excel, and Sigma plot, and the statistical analysis was performed by ANOVA and Fisher’s LSD. The *p*-value was set at 0.05.

## 4. Conclusions

From 23 candidate LAB strains, we identified *L. casei* BCRC10397 as a suitable strain for djulis sourdough fermentation. This strain was selected for its ability to decrease the pH level to 4.6 and increase proteolysis capacity by 1.88 times (relative to controls). Optimal fermentation parameters obtained via RSM significantly improved the antioxidant activity, TEAC, total phenols, total flavonoids, and total peptides. DSB groups decreased the specific volume of the bread while increasing its hardness and chewiness. Therefore, considering the effect of the texture and nutritional improvement, 20% djulis was added to produce bread. The DSB not only provided great functionality and extended the shelf life of the bread, but it also improved its acceptability. The addition and fermentation of djulis shows its great potential in relation to functional foods. Thus, large-scale production of DSB, determination of the involved antioxidant mechanisms, and identification of the bioactive components are valuable directions for future research.

## Figures and Tables

**Figure 1 molecules-26-05658-f001:**
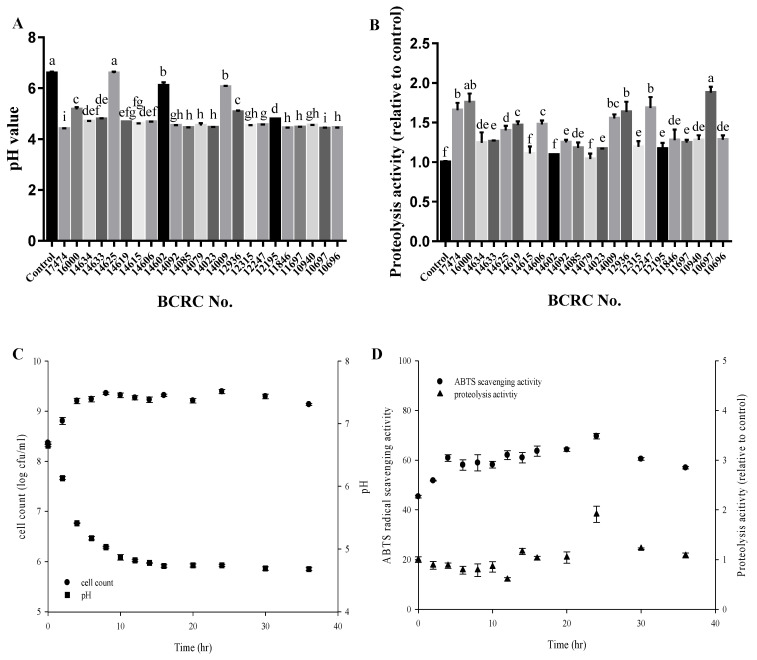
The selection of different lactic acid bacteria for djulis fermentation through the determination of (**A**) the pH value and (**B**) proteolysis activity of djulis fermented by different LAB strains. (**C**) Growth curve and (**D**) ABTS radical scavenging activity and proteolysis activity of djulis fermented by *L. casei* BCRC 10697. Values are the mean ± SD. Superscripts (a, b, c, etc.) indicate significant differences in Fisher’s LSD tests (*p* < 0.05).

**Figure 2 molecules-26-05658-f002:**
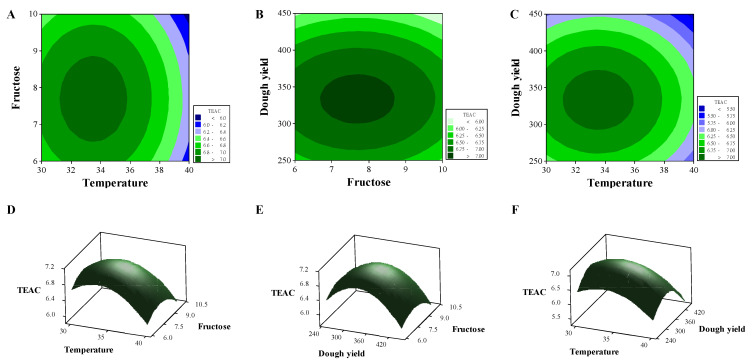
Optimization of conditions for TEAC of sourdough fermentation using response surface methodology (RSM). Contour plots of (**A**) fructose vs. temperature, (**B**) dough yield vs. fructose, and (**C**) dough yield vs. temperature. Surface plots (**D**–**F**) correspond to (**A**–**C**), respectively.

**Figure 3 molecules-26-05658-f003:**
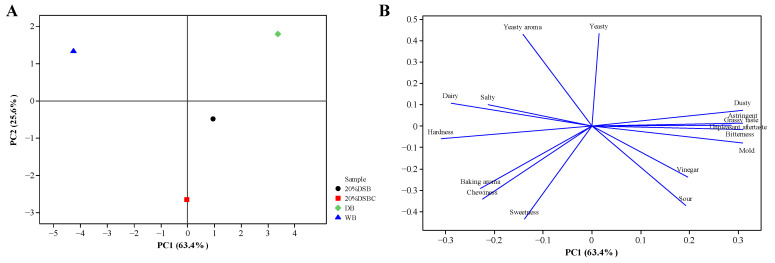
Descriptive analysis of different bread samples. (**A**) Score plot and (**B**) loading plot of different bread samples. PC1 explained 63.4% of the variation in the data. WB (wheat bread) is marked in blue, DB (djulis bread) in green, 20% DSB (djulis sourdough bread) in black, 20% DSBC (djulis sourdough bread with cocoa power and dried cranberry) in red, and attributes are marked in blue.

**Table 1 molecules-26-05658-t001:** Experimental design for RSM of the bread samples.

Run	Temperature (°C, *X*_1_)	Fructose (%, *X*_2_)	Dough Yield (*X*_3_)	TEAC (mmol/kg, *Y*)
1	35	10	250	6.18
2	40	6	350	6.24
3	35	8	350	6.98
4	35	8	350	6.92
5	35	6	450	6.04
6	30	8	450	6.04
7	30	10	350	6.42
8	35	6	250	6.32
9	35	10	450	5.75
10	30	6	350	6.45
11	35	8	350	6.98
12	40	8	450	5.19
13	40	8	250	5.77
14	40	10	350	5.98
15	30	8	250	6.55

**Table 2 molecules-26-05658-t002:** ANOVA of the response variables of the bread samples.

Source	DF	Adj MS	*F*-Value	*p*
Regression	9	0.41167	9.99	0.010 **
Linear	3	0.44755	10.86	0.013 *
Temperature	1	0.75537	18.33	0.008 **
Fructose	1	0.23844	5.78	0.061
Dough yield	1	0.78291	18.99	0.007 **
Square	3	0.85519	20.75	0.003 **
Temperature × Temperature	1	0.85796	20.82	0.006 **
Fructose × Fructose	1	0.32087	7.78	0.038 *
Dough yield × Dough yield	1	1.71256	41.55	0.001 ***
Interaction	3	0.00655	0.16	0.919
Temperature × Fructose	1	0.01254	0.30	0.605
Temperature × Dough yield	1	0.00119	0.03	0.872
Fructose × Dough yield	1	0.00593	0.14	0.720
Residual Error	5	0.04122		
Lack-of-Fit	3	0.04748	1.49	0.425 > 0.05
Pure Error	2	0.03183		
Total	14			

DF = degrees of freedom, Adj MS = adjusted mean square, * = *p* ≤ 0.05, ** = *p* ≤ 0.01, *** = *p* ≤ 0.001.

**Table 3 molecules-26-05658-t003:** Bioactive content of unfermented, fermented, and optimally fermented products.

Groups	TEAC_ABTS_ (mmol/kg)	TEAC_DPPH_ (mmol/kg)	Total Phenolic Compounds (mg GAE/g)	Total Flavonoid Compounds (mg QE/g)	Total Peptides (mg leucine/g)
Unfermented grains	4.12 ± 0.11 ^a^	4.55 ± 0.21 ^b^	4.62 ± 0.20 ^b^	2.14 ± 0.05 ^b^	1.78 ± 0.11 ^c^
Original product	6.23 ± 0.13 ^b^	5.04 ± 0.12 ^a^	6.14 ± 0.38 ^a^	2.44 ± 0.07 ^a^	3.20 ± 0.10 ^b^
Optimal product	6.82 ± 0.085 ^a^	5.63 ± 0.25 ^a^	6.88 ± 0.1 ^a^	2.50 ± 0.07 ^a^	3.81 ± 0.26 ^a^

Statistical differences were calculated using ANOVA and Fisher’s LSD. Values are the mean ± SD of three independent experiments. Superscripts (a, b, c) indicate significant differences (*p* < 0.05).

**Table 4 molecules-26-05658-t004:** Bioactive content of the different bread samples.

Groups	TEAC_ABTS_ (mmol/kg)	TEAC_DPPH_ (mmol/kg)	Total Phenolic Compounds (mg GAE/g)	Total Flavonoid Compounds (mg QE/g)	Total Peptides (mg leucine/g)
WB	1.79 ± 0.30 ^c^	0.02 ± 0.02 ^d^	2.01 ± 0.15 ^c^	0.05 ± 0.002 ^c^	1.23 ± 0.07 ^c^
DB	3.09 ± 0.03 ^b^	0.73 ± 0.02 ^c^	3.59 ± 0.13 ^b^	0.47 ± 0.01 ^b^	2.48 ± 0.19 ^b^
20% DSB	3.55 ± 0.10 ^a^	0.89 ± 0.07 ^b^	4.23 ± 0.13 ^a^	0.55 ± 0.01 ^a^	3.47 ± 0.05 ^a^
25% DSB	3.46 ± 0.08 ^a^	1.12 ± 0.03 ^a^	4.22 ± 0.22 ^a^	0.58 ± 0.04 ^a^	3.35 ± 0.12 ^a^

Statistical differences were calculated using ANOVA and Fisher’s LSD. Values are the mean ± SD of three independent experiments. Superscripts (a, b, c, d) indicate significant differences (*p* < 0.05).

**Table 5 molecules-26-05658-t005:** Texture profile analysis and proximate analysis of the different bread samples.

Characteristics	WB	DB	20% DSB	25% DSB
Specific volume (cm^3^/g)	3.91 ± 0.65 ^a^	3.43 ± 0.62 ^b^	2.55 ± 0.49 ^c^	2.67 ± 1.83 ^c^
Area (%)	50.49 ± 0.65 ^a^	39.42 ± 0.62 ^c^	44.41 ± 0.49 ^b^	42.13 ± 1.83 ^b^
Particle size (mm^2^)	0.20 ± 0.10 ^b^	0.50 ± 0.00 ^a^	0.30 ± 0.10 ^a^	0.40 ± 0.00 ^a^
Hardness (g)	983.10 ± 61.57 ^c^	973.90 ± 113.10 ^c^	1789.00 ± 232.70 ^b^	3121.00 ± 96.81 ^a^
Springiness	1.03 ± 0.05 ^a^	0.99 ± 0.003 ^a^	0.9813 ± 0.004 ^a^	0.9717 ± 0.001 ^a^
Cohesiveness	0.002 ± 0.90 ^a^	0.001 ± 0.90 ^a^	0.003 ± 0.88 ^b^	0.002 ± 0.87 ^b^
Chewiness (g)	919.10 ± 96.46 ^c^	865.30 ± 99.80 ^c^	1546.00 ± 189.00 ^b^	2632.00 ± 87.96 ^a^
Resilience	0.59 ± 0.01 ^a^	0.56 ± 0.004 ^b^	0.54 ± 0.009 ^b,c^	0.51 ± 0.002 ^d^
Moisture (%)	35.16 ± 0.76	33.35 ± 0.60	32.52 ± 0.10	30.33 ± 0.20
Fat (%)	1.05 ± 0.84	1.31 ± 1.1	2.11 ± 1.71	2.16 ± 1.93
Protein (%)	8.99 ± 0.11	9.89 ± 0.06	10.39 ± 0.10	10.84 ± 0.15
Carbohydrate (%)	53.91 ± 0.98	54.11 ± 0.53	53.98 ± 0.42	55.23 ± 0.20
Ash (%)	1.10 ± 0.06	1.55 ± 0.05	1.41 ± 0.03	1.68 ± 0.09

Statistical differences were calculated using ANOVA and Fisher’s LSD. Values are the mean ± SD of three independent experiments. Superscripts (a, b, c, d) indicate significant differences (*p* < 0.05).

**Table 6 molecules-26-05658-t006:** Affective analysis of the different bread samples.

Groups	Appearance	Flavor	Taste	Texture	Overall Acceptability
WB	6.20 ± 0.28 ^a,b^	5.89 ± 0.28 ^b^	5.77 ± 0.29 ^a,b^	5.89 ± 0.27 ^a,b^	6.00 ± 0.26 ^a,b^
DB	6.06 ± 0.23 ^b^	5.20 ± 0.21 ^c^	4.40 ± 0.26 ^b^	5.03 ± 0.33 ^c^	4.74 ± 0.27 ^b^
20%DSB	6.11 ± 0.25 ^b^	5.74 ± 0.23 ^b,c^	5.11 ± 0.27 ^b^	5.71 ± 0.29 ^a,b,c^	5.31 ± 0.23 ^b^
20%DSBC	6.89 ± 0.24 ^a^	6.57 ± 0.21 ^a^	6.09 ± 0.26 ^a^	6.46 ± 0.23 ^a^	6.31 ± 0.23 ^a^

Statistical differences were calculated using ANOVA and Fisher’s LSD. Values are the mean ± SD of three independent experiments. Superscripts (a, b, c) indicate significant differences (*p* < 0.05).

**Table 7 molecules-26-05658-t007:** Recipe of breads.

Type (%)	WB	DB	20% DSB	25% DSB	DSB with 20% Cocoa Powder
**Djulis sourdough**					
djulis	-	-	20	25	20
water			46.6	58.25	46.6
**Bread dough**					
Wheat flour	100	80	80	75	80
Djulis flour	-	20	-	-	-
Water	60	60	13.4	1.75	13.4
Sugar	5	5	5	5	5
Salt	1	1	1	1	1
Yeast	2	2	2	2	2
Oil	5	5	5	5	5
Cocoa powder					5
Dried cranberry					20

The total weight of each group was 100 g.

## Data Availability

The data presented in this study are available on request from the corresponding author.

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
