# Peer review of "Development and Optimization of Djulis Sourdough Bread Fermented by Lactic Acid Bacteria for Antioxidant Capacity"

_molecules, 2021, doi:10.3390/molecules26185658_

Round 1

Reviewer 1 Report

Dear authors your manuscript requires significant language editing as my review in many points was done by significant level of assumptions from my side. As this is not an appropriate way of reviewing I would like to ask you to review and make all appropriate changes in the syntax and grammar of the manuscript as your manuscript cannot yet be fully reviewed.

Author Response

Reviewer #1:

Dear authors your manuscript requires significant language editing as my review in many points was done by significant level of assumptions from my side. As this is not an appropriate way of reviewing I would like to ask you to review and make all appropriate changes in the syntax and grammar of the manuscript as your manuscript cannot yet be fully reviewed.

Thank you very much for your precious comments. We revised our manuscript through the English editing service.

Reviewer 2 Report

In their work, the authors report the optimization process [Box Benken design-RSM; factors: fermentation time (FT), fructose concentration (FC), and dough yield (DY)] of a Djulis (Chenopodium formosanum) based-sourdough bread fermented with LAB. The most effective proteolytic strain (Peakmax 24h) was Lactobacillus casei BCRC 10697 (pH drop from 6.7 to 4.8, ~9.3 logCFU from 5 to >30h), enhancing sourdough´s radical scavenging activity (60-73% from 5-25h). The highest TEAC value (t.12 mmol/kg) was achieved at 33.5oC (FT), 7.7% (FC) and 332.8 (DY). Total peptides>phenolic compounds were released in the optimized sourdough when compared to unfermented Djulis grains. Bread density, hardness, and chewiness were particularly affected by partial substitution with Djulis in a dose-dependent manner. 20% and 25% wheat flour substitution with Djulis did not affect the sensory profile of final bread, although dry cranberry and cocoa addition further increases the consumer acceptability of 20% substituted bread.

The study presents relevant information, but minor aspects are suggested to improve the scientific soundness and contribution of this study:

  • The readability and syntax of the manuscript will be substantially improved if it is reviewed by a formal translation agency or by a native English spoken person.
  • Title. OK.  
  • Abstract. It should be more concise and quantitative without sacrificing important differential results.
  • Introduction & conclusion. The authors should further highlight the fact that you use alternative sources of grains (particularly pseudo-cereals) to improve the nutritional profile (e.g., digestible protein), functional (e.g., reduced allergenicity, gluten-free, antioxidant capacity) of wheat breads (DOI: 1016/j.tifs.2018.08.015 , 10.1016/j.tifs.2018.03.016 , 10.9755/ejfa.2020.v32.i9.2145 10.9755/ejfa.2020.v32.i9.2145 , 10.3389/fnut.2019.00098). Do the same with the conclusion section.
  • Results & discussion. The authors should start this section with a brief introduction to the technological and nutritional importance of Djulis flour and the advantages of its fermentation in improving the sensory profile of baked goods.
  • Tables (T) & Figures (F). Please improve resolution of almost all figures (300 dpi or more) and all figures must be extensively explained, otherwise include as supplementary material. F1a, b: Arrange experimental treatments from highest to lowest value and include treatment F1c,d: It is suggested to eliminate the lines that connect the points and include the equation and line that best explain the behavior of these points (goodness-of-fit technique). Please standardize the tables according to the Molecules guidelines. The results in Table 6 could be transformed into spider diagrams, as is customary to report in sensory analysis. Also, if possible, photographs of the loaves that illustrate the findings reported in this table would be highly recommended.

References. OK

Author Response

Reviewer #2:

In their work, the authors report the optimization process [Box Benken design-RSM; factors: fermentation time (FT), fructose concentration (FC), and dough yield (DY)] of a Djulis (Chenopodium formosanum) based-sourdough bread fermented with LAB. The most effective proteolytic strain (Peakmax 24h) was Lactobacillus casei BCRC 10697 (pH drop from 6.7 to 4.8, ~9.3 logCFU from 5 to >30h), enhancing sourdough´s radical scavenging activity (60-73% from 5-25h). The highest TEAC value (t.12 mmol/kg) was achieved at 33.5oC (FT), 7.7% (FC) and 332.8 (DY). Total peptides>phenolic compounds were released in the optimized sourdough when compared to unfermented Djulis grains. Bread density, hardness, and chewiness were particularly affected by partial substitution with Djulis in a dose-dependent manner. 20% and 25% wheat flour substitution with Djulis did not affect the sensory profile of final bread, although dry cranberry and cocoa addition further increases the consumer acceptability of 20% substituted bread.

Thank you very much for your precious comments. We replied them item by item carefully as follows:

#1 The readability and syntax of the manuscript will be substantially improved if it is reviewed by a formal translation agency or by a native English spoken person.

We revised our manuscript through the English editing service.

#2 Abstract. It should be more concise and quantitative without sacrificing important differential results.

We rewrote Abstract (216 words) in Line 19 in the revised manuscript as follows:

This study developed a nutritionally valuable product with bioactive activity that improves the quality of bread. Djulis (Chenopodium formosanum), a native plant of Taiwan, was fermented using 23 different lactic acid bacteria strains. Lactobacillus casei BCRC 10697 was identified as the ideal strain for fermentation, as it lowered the pH value of samples to 4.6 and demonstrated proteolysis ability 1.88 times higher than controls after 24 hours of fermentation. Response surface methodology was adopted to optimize the djulis fermentation conditions for trolox equivalent antioxidant ca-pacity (TEAC). The optimal conditions were a temperature of 33.5°CC, fructose content of 7.7%, and dough yield of 332.8, yielded a TEAC at 6.82 mmole/kg. A 63% increase in TEAC and 20% increase in DPPH were observed when compared with unfermented djulis. Subsequently, the fermented djulis was used in different proportions as a substitute for wheat flour to make bread. The total phenolic and flavonoid compounds were 4.23 mg GAE/g and 3.46 mg QE/g, marking re-spective increases of 18% and 40% when the djulis was added. Texture analysis revealed that adding djulis increased the hardness and chewiness of sourdough breads. It also extended their shelf life by approximately 2 days. Thus, adding djulis to sourdough can enhance the functionality of breads and may provide a potential solution for developing djulis-based functional food.

#3 Introduction & conclusion. The authors should further highlight the fact that you use alternative sources of grains (particularly pseudo-cereals) to improve the nutritional profile (e.g., digestible protein), functional (e.g., reduced allergenicity, gluten-free, antioxidant capacity) of wheat breads (DOI: 1016/j.tifs.2018.08.015 , 10.1016/j.tifs.2018.03.016 , 10.9755/ejfa.2020.v32.i9.2145 10.9755/ejfa.2020.v32.i9.2145 , 10.3389/fnut.2019.00098). Do the same with the conclusion section.

        We added description about the promotion of wheat breads by using alternative sources of grains in Line 42-46, and cited the references in the revised manuscript as follows:

For instance, …. Adding quinoa in particular also lowers blood lipids and increases an-tioxidant capability [4]. Moreover, alternative sources of cereals, especially pseudo-grains, improve the nutritional and functional components of traditional wheat bread[5], included increasing the content of digestible protein[6], reducing allergies by making gluten-free bread[7], and enhancing antioxidant capacity[8].

[5] Tebben, L.; Shen, Y.; Li, Y., Improvers and functional ingredients in whole wheat bread: A review of their effects on dough properties and bread quality. Trends in Food Science & Technology 2018, 81, 10-24. doi: https://doi.org/10.1016/j.tifs.2018.08.015

[6] Rollán, G. C.; Gerez, C. L.; LeBlanc, J. G., Lactic Fermentation as a Strategy to Improve the Nutritional and Functional Values of Pseudocereals. Frontiers in Nutrition 2019, 6, (98). doi: https://doi.org/10.3389/fnut.2019.00098

[7] Toth, M.; Vatai, G.; Koris, A., Gluten-Free Bread from ingredients and nutrition point of view: A Mini-Review. Emirates Journal of Food and Agriculture 2020, 32, 634. doi: 10.9755/ejfa.2020.v32.i9.2145

[8] Mir, N.; Riar, C. s.; Singh, S., Nutritional constituents of pseudo cereals and their potential use in food systems: A review. Trends in Food Science & Technology 2018, 75. doi: 10.1016/j.tifs.2018.03.016

#4 Results & discussion. The authors should start this section with a brief introduction to the technological and nutritional importance of Djulis flour and the advantages of its fermentation in improving the sensory profile of baked goods.

        We added the brief introduction of Djulis in the Results & discussion section in Line 92-98 as follows:

Due to increasing health concerns, …. Studies have confirmed that sourdough fermentation by LAB enhances bioactive capacity and improves texture. The nutritional content of djulis was rich in protein, lipid, ash and essential amino acids, which was particularly rich in amino acids with sulfur groups and lysine, and it was one of the few vegetable proteins that provided complete essential amino acids[27]. Among djulis fermentation, the lactic acid bacteria (LAB) promoted the acidification of dough and the hydrolysis of protein[28], the decrease in pH value extended the shelf life of bread[29], and the produced protease and reductase weakened the gluten structure and make the bread texture softer[30].

[27] Filho, A. M.; Pirozi, M. R.; Borges, J. T.; Pinheiro Sant'Ana, H. M.; Chaves, J. B.; Coimbra, J. S., Quinoa: Nutritional, functional, and antinutritional aspects. Critical reviews in food science and nutrition 2017, 57, (8), 1618-1630.

[28] Liu, T.; Li, Y.; Chen, J.; Sadiq, F.; Zhang, G.; Li, Y.; He, G.-q., Prevalence and diversity of lactic acid bacteria in Chinese traditional sourdough revealed by culture dependent and pyrosequencing approaches. Lwt - Food Science and Technology 2016, 68, 91-97.

[29] Settanni, L., Sourdough and cereal-based foods: Traditional and innovative products. 2017; pp 199-230.

[30] Nutter, J.; Fritz, R.; Saiz, A. I.; Iurlina, M. O., Effect of honey supplementation on sourdough: Lactic acid bacterial performance and gluten microstructure. LWT 2017, 77, 119-125.

#5 Tables (T) & Figures (F). Please improve resolution of almost all figures (300 dpi or more) and all figures must be extensively explained, otherwise include as supplementary material. F1a, b: Arrange experimental treatments from highest to lowest value and include treatment F1c,d: It is suggested to eliminate the lines that connect the points and include the equation and line that best explain the behavior of these points (goodness-of-fit technique). Please standardize the tables according to the Molecules guidelines. The results in Table 6 could be transformed into spider diagrams, as is customary to report in sensory analysis. Also, if possible, photographs of the loaves that illustrate the findings reported in this table would be highly recommended.

We improved the resolution of almost all figures and standardize tables according to the Molecules guidelines. Also, we made new explanation in our manuscript as follows:

Figure 1: (line 114-118)

The selection of different lactic acid bacteria for djulis fermentation through the determination of(A) The pH value and (B) proteolysis activity of djulis fermented by different LAB strains. (C) Growth curve and (D) ABTS radical scavenging activity and proteolysis activity of djulis fermented by L. casei BCRC 10697. Values are the mean ± SD. Superscripts (a, b, c) indicate significant differences in Fisher’s LSD tests (p < 0.05).

Figure 2: (line 149-151)

Optimization of conditions for TEAC of sourdough fermentation using response surface methodology (RSM). Contour plots of (A) fructose vs. temperature, (B) dough yield vs. fructose, and (C) dough yield vs. temperature. Surface plots (D), (E), and (F) correspond to (A), (B), and (C), respectively

Figure 3: (line 152-153)

Descriptive analysis of different bread samples. (A) Score plot and (B) Loading plot of different bread samples. PC1 explained 63.4% of the variation in the data. WB (wheat bread) was marked in blue, DB (djulis bread) in green, 20% DSB (djulis sourdough bread) in black, 20% DSBC (djulis sourdough bread with cocoa power and dried cranberry) in red, and attributes were marked in blue.

Table 6 was considering that there were not as many variants in texture profile analysis as sensory evaluation. Therefore, if we apply principal component analysis to the texture characteristics of bread, we may lose some information and lead to poor results.

Reference: Geiger, B. C., & Kubin, G. (2012, September). Relative information loss in the PCA. In 2012 IEEE Information Theory Workshop (pp. 562-566). IEEE.

Reviewer 3 Report

Dear Author:

I think the manuscript entitled “Development and Optimization of Djulis Sourdough Bread Fermented by Lactic Acid Bacteria for Antioxidant Capacity” possess novelty as it deals with a no so well documented yet specific bread “Djulis Sourdough” fermented by lactic acid while most bread are fermented by yeasts. The quality parameter Antioxidant Capacity is quite interesting. I think that the material and methods are adequate, results are properly discussed and the graphs are of good quality.

Best Regards.

Author Response

Reviewer #3:

I think the manuscript entitled “Development and Optimization of Djulis Sourdough Bread Fermented by Lactic Acid Bacteria for Antioxidant Capacity” possess novelty as it deals with a no so well documented yet specific bread “Djulis Sourdough” fermented by lactic acid while most bread are fermented by yeasts. The quality parameter Antioxidant Capacity is quite interesting. I think that the material and methods are adequate, results are properly discussed and the graphs are of good quality.

        Thank you very much for your precious comments.

Round 2

Reviewer 2 Report

Thank you very much for accepting most of my suggestions. Well done